# Task Allocation Model Based on Worker Friend Relationship for Mobile Crowdsourcing

**DOI:** 10.3390/s19040921

**Published:** 2019-02-22

**Authors:** Bingxu Zhao, Yingjie Wang, Yingshu Li, Yang Gao, Xiangrong Tong

**Affiliations:** 1School of Computer and Control Engineering, Yantai University, Yantai 264005, China; zbx_ytu@163.com (B.Z.); togaoyang@163.com (Y.G.); txr@ytu.edu.cn (X.T.); 2Department of Computer Science, Georgia State University, Atlanta, GA 30303, USA

**Keywords:** mobile crowdsourcing, task allocation, social networks, GeoHash

## Abstract

With the rapid development of mobile devices, mobile crowdsourcing has become an important research focus. According to the task allocation, scholars have proposed many methods. However, few works discuss combining social networks and mobile crowdsourcing. To maximize the utilities of mobile crowdsourcing system, this paper proposes a task allocation model considering the attributes of social networks for mobile crowdsourcing system. Starting from the homogeneity of human beings, the relationship between friends in social networks is applied to mobile crowdsourcing system. A task allocation algorithm based on the friend relationships is proposed. The GeoHash coding mechanism is adopted in the process of calculating the strength of worker relationship, which effectively protects the location privacy of workers. Utilizing synthetic dataset and the real-world Yelp dataset, the performance of the proposed task allocation model was evaluated. Through comparison experiments, the effectiveness and applicability of the proposed allocation mechanism were verified.

## 1. Introduction

In recent years, a new framework has emerged, and mobile crowdsourcing has enable workers to perform spatiotemporal tasks. With the development of various sensors (such as GPS and TP (Temperature Probe)) and wireless mobile networks (such as 4G and 5G), the ability of smart devices to process data is getting stronger and stronger. It is easy to participate in mobile crowdsourcing for performing spatial tasks at specific locations for people, such as taking photos/videos [1], uploading traffic information [2], weather conditions, online courier picking, booking services and dating/meeting apps and games (e.g., Pokémon Go). It has greatly improved the quality of life and promoted the development of online technology. At the same time, the market research institute Newzoo [3] recently released the “2018 Global Mobile Market Report”, showing that by the end of 2018, the number of global smartphone users reached 3.3 billion, which means that there will be many users in the future mobile crowdsourcing system. In general, mobile crowdsourcing systems (e.g., drip taxis [4], hungry apps [5], etc.) can allocate or recommend spatial tasks to active workers who are close to the task location. The mobile crowdsourcing system allocates the most suitable crowd workers to each task based on the spatiotemporal attributes and the characteristics of tasks.

The most striking difference between mobile crowdsourcing and traditional crowdsourcing is that workers can only perform the part of the spatial task close enough to them, so that workers can actually move to the spatial task’s location before the task deadline. However, workers have no spatial restrictions in traditional crowdsourcing systems. Therefore, researching and designing an effective task allocation model to allocate tasks for workers is the main goal of mobile crowdsourcing.

In [6], tasks of mobile crowdsourcing systems can be published in two different modes: Worker Selected Tasks (WST) and Server Allocated Tasks (SAT). For WST mode, online workers can choose any spatial task published by nearby requesters without having to coordinate with the mobile crowdsourcing server. Due to the autonomy of WST, the platform cannot control the workers to achieve a global optimal allocation strategy in terms of the number of allocated tasks or some objective scores. For SAT mode, online workers will periodically report their location to the platform, so that the platform can allocate tasks to nearby workers to optimize the effectiveness of mobile crowdsourcing. In WST mode, workers do not need to report their location to the platform, but in SAT mode the platform needs to track the locations of workers and protect the spatial privacy of workers. In addition, in WST mode, certain tasks may never be selected because the platform has no control over the characteristics of workers. However, in SAT mode, the server can allocate these tasks to the workers to maximize the number of tasks allocated or maximize platform’s utility [7]. Therefore, in our paper, we propose a task allocation model in SAT mode. Although there are many studies [7,8,9,10,11,12] on task allocation methods for mobile crowdsourcing, their problem settings and optimization goals are not exactly same.

At present, few research works [7,10,11,12,13] considered the relationship between friends when allocating task in mobile crowdsourcing. A study [14] in the journal Nature Communications pointed out that from the brain scan results, when friends react to the same thing, the brain wavelengths are extremely similar. The research team led by Carolyn Parkinson, a social psychologist at the University of California, Los Angeles, conducted experiments on 42 students. The researchers carried out each of the reaction images one by one. In comparison, brain activity patterns are used to predict who is a friend and who is a classmate. The final correct rate reached 48%. In the final report, the researchers wrote: “Neural similarity increases with the friendship between people. These results show that we are very similar to our friends in the way we perceive and respond to the world around us”. At the same time, the reaction between friends is more similar than the reaction between non-friends. The more similar the responses are, the more intimate their relationship are. Therefore, this paper deeply explores the relationship between people and their friends to allocate the same or similar tasks that the worker has performed to the workers’ friends, or allocate common tasks to the workers and his friends, thereby reducing costs. For example, car pooling behavior saves more money than special car behavior. The corresponding task/worker categories are shown in Table 1.

**Example.** In Table 1, A–F represent six workers, while 1–5 represent five tasks, where Task 1 and Task 5 belong to the same category of task. As shown in Figure 1, we assume that Worker A and Worker B are university roommates who live together everyday. They are intimate, and not only have similar professional skills, but are familiar with each other and understand each other. Completing a job will be more tacit.

In SAT mode, the mobile crowdsourcing platform can allocate Task 5 to Worker A and Worker B, so that the two workers can work together. It will not only improve the quality of sensed data and the completion rate of tasks, but also save the cost of moving to the target location, which can reduce the unnecessary overhead and maximize the use of current resources. Conversely, if Task 5 is allocated to Worker E, Worker E will be required to bear the cost of task alone. For a more specific example, if Workers A and B are classmates with the same major, they have similar professional skills. We assume that Workers A and B are majoring in computer science and technology. The five-pointed star represents the task that needs ability of repairing computers. Worker A has performed Task 5. The mobile crowdsourcing platform can allocate Task 1 to Workers A and B who have similar job skills. The effect will be perfectly reflected in the completion, professionalism, time saving and allocation matching of the tasks.

According to the above analysis, in this paper, we propose a Friend Relationship Strength Mobile Crowdsourcing (FRS-MC) combining the adaptive threshold algorithm and GeoHash coding, which aims to deeply explore the strength of the relationship between workers and their friends. Under the constraints of the deadline of task and budget, this paper applies friend relationships to the task allocation problem of mobile crowdsourcing, which can improve the accuracy of task allocation, reduce the travel cost of workers and maximize the total utilities of the allocations (defined as the total utilities of workers). Because the platform needs to track the locations of workers in SAT mode, it needs to protect the spatial privacy of workers. GeoHash coding can provide high protection of worker’s spatial location information. First, we explore the relationship strength between workers and their friends on time attribute. Then, we explore the relationship strength between workers and their friends on geographical attributes. Finally, our proposed FRS-MC algorithm is used to select the workers with the highest relationship strength.

Specifically, we make the following contributions.
We research the relationships among friends in social networks and apply them to task allocation model in mobile crowdsourcing.The time relationship strength based on historical interaction information of friends, and the geographical relationship strength based on GeoHash coding are researched to protect the privacy of workers.We design an algorithm for task allocation based on friend relationship strength.Utilizing real dataset, the adaptation and effectiveness of the proposed FRS-MC algorithm are verified through comparison with other task allocation methods.

In Section 2, we review the previous research in mobile crowdsourcing. Section 3 introduces the definition of problems in the mobile crowdsourcing system. In Section 4, we introduce the proposed model and give the algorithm. Finally, Section 6 summarizes the paper.

## 2. Related Work

In this section, we review the related works of mobile crowdsourcing, as well as task allocation issues and incentives.

### 2.1. Mobile Crowdsourcing

Mobile crowdsourcing is one of the emerging research focuses in resent years. It has attracted great attentions from practitioners and scholars for many years. Anastasios et al. [15] researched the meaningful spatiotemporal patterns and user’s mobility by analyzing users’ sign-in behaviors. Howe et al. [16] first proposed the concept of crowdsourcing. Many systems and applications have promoted the development of mobile crowdsourcing; e.g., SensoRcivico [17] provided a common platform to support a variety of crowdsourcing tasks. Hu et al. [18] utilized crowdsourcing to improve the quality of Point of Interest (POI) labels. However, thus far, few studies have stated what has been achieved and what should be done. Zhao et al. [19] critically studied the basis of crowdsourcing research by investigating the landscape of existing research, including theoretical foundations, research methods and research focus.

Kazemi et al. [6] classified mobile crowdsourcing systems from two perspectives: people’s motivation and publishing model. From the perspective of people’s motivation, the crowd workers can be divided into two categories: reward-based workers can be rewarded according to the services they provide; and self-based workers voluntarily perform tasks. In this paper, we propose FRS-MC algorithm based on reward model that workers get paid after completing tasks.

### 2.2. Task Allocation

According to the release mode of crowd tasks, mobile crowdsourcing can also be divided into two categories: worker selection task (WST) and server allocation task (SAT) [6]. In particular, for the WST model, crowd tasks are published to all workers, and workers can choose any task themselves. However, maximizing social welfare in task allocation is a NP-hard problem. Cheung et al. [20] proposed an Asynchronous and Distributed Task Selection (ADTS) algorithm to help workers select tasks. Conversely, for SAT mode, the mobile crowdsourcing platform will allocate tasks directly to workers based on the location information.

Some of the previous works for WST model allow workers to select available tasks based on their personal preferences [21,22]. However, according to the SAT mode, tasks in mobile crowdsourcing system are allocated to workers for different purposes. For example, Hassan et al. [8], To et al. [9], and Kazemi et al. [6] considered the problem of online spatial task allocation; the main goal is to maximize the allocation for all tasks. Luan et al. [10] introduced hyperlocal mobile crowdsourcing, and To et al. [11] maximized the number of allocated tasks with budget constraints. Based on the crowdsourcing methods, Zhai et al. [23] studied the distributed scheme and allocated spectrum sensing tasks for mobile terminals. The goal is to maximize the perceptual effect function to solve the task allocation problem in cognitive radio networks. Li et al. [24] considered dynamic participant recruitment issues for heterogeneous sensing tasks (with different time and spacial requirements) to minimize sensing costs while maintaining a certain level of probability coverage. Dang et al. [7] arranged the largest number of complex tasks for crowd workers based on worker and task constraints, and minimized the cost of travel for allocation, while guaranteeing the maximal number of task allocations.

Tong et al. [25] solved the shortcomings of the existing offline methods (micro-tasks and mass workers in actual applications appear dynamically and their spatiotemporal information cannot be known in advance). Most previous works on mobile crowdsourcing design task allocation strategies to maximize allocated scores. However, these methods are designed based on the available workers/tasks in mobile crowdsourcing system when the worker/task is allocated. These strategies may achieve local optimum due to ignoring future workers/tasks that may join the system. Cheng et al. [26] considered both existing and future (through prediction) workers/tasks to achieve global optimum task allocations. Lian et al. [27] considered a reliable diversity-based mobile crowdsourcing problem, and allocated workers for spatial tasks to maximize the diversity and reliability of mobile crowdsourcing system. Cheng et al. [12] considered a mobile crowdsourcing scenario that each worker has a set of qualified skills, and each spatial task (e.g., repairing a house, decorating a room, and entertaining a ceremony) is time-limited and budget-limited.

### 2.3. Other Related Works

In traditional crowdsourcing, crowd participants accept and complete tasks through the web platform without having to consider privacy protection issues. In mobile crowdsourcing, crowdsourcing participants need to submit location information to the platform, which generates the risk of privacy leakage. The location information of crowdsourcing participants is also an important factor to be considered for task allocation. The location privacy protection problem has been studied in the field of location-based services. Wang et al. [28] proposed an incentive mechanism for static selection of worker candidates, and then dynamically selected the winner after bidding. Under the framework of this incentive mechanism, privacy protection is proposed to protect the privacy of workers. Wang et al. [29,30] proposed a real-time location privacy protection for mobile crowdsourcing systems. An improved two-stage auction algorithm based on trust and privacy sensitivity (TATP) is proposed. They also proposed a k-ε-differential privacy protection to prevent user’s location information from leaking. Chi et al. [31] proposed a location privacy protection mechanism CKD (k-anonymity and differential privacy-preserving). Hu et al. [32] proposed an incentive mechanism based on multi-attribute reverse auction for mobile crowdsourcing. Li et al. [33] proposed two algorithms to select appropriate mobile crowdsensing participants and calculate the payments to them. Duan et al. [34] proposed two distributed auction schemes, namely cost-preferred auction scheme (CPAS) and time schedule-preferred auction scheme (TPAS), which differ on the methods of task scheduling, winner determination, and payment computation. Li et al. [35] proposed a novel network model and an influence propagation model taking influence propagation in both online social networks and the physical world into consideration. Cai et al. [36] proposed a novel mechanism for data uploading in smart cyber-physical systems, which considers both energy conservation and privacy preservation. Zhang et al. [37] proposed a novel privacy checking algorithm and an efficient one to accelerate the privacy checking process. Liu et al. [13,38] constructed a security protocol that preserves the location privacy of workers and task requesters. To et al. [39] distributed the location of tasks and workers based on geographic indistinguishability, and then designed techniques to quantify the achievable probabilities between tasks and workers.

## 3. Problem Definition

In this section, we present the definitions of the task allocation model based on friend relationship for mobile crowdsourcing.

### 3.1. Crowdsourcing Workers with Friends

First, we give the definition of crowdsourcing workers with friends in mobile crowdsourcing. Assume that $P_{t}=\left\{p_{1},p_{2},\cdots ,p_{k}\right\}$ is a set of all workers in timestamp *t* in mobile crowdsourcing, that is, there are *k* workers at timestamp *t*. Each of them can perform some types of crowdsourcing tasks.

**Definition** **1.**
*(Crowdsourcing Workers with Friends) $W=\left\{w_{1},w_{2},\ldots ,w_{m}\right\}$ represents the set of crowdsourcing workers in the mobile crowdsourcing, each $w_{i}\left(1\leq i\leq n\right)$ has a set of friends $\Omega_{i}=\left\{f_{1},f_{2},\ldots ,f_{r}\right\}\left(\in P_{t}\right)$, which means that $w_{i}$ has r friends. The crowdsourcing worker can move with the speed $v_{i}$ and has a maximal moving distance $d_{i}$. The service quality of $w_{i}$ is presented by $q_{i}$.*


In Definition 1, crowd workers can move dynamically with the speed $v_{i}$ in any direction, the position of $w_{i}$ is shown by $l_{i}$.They can freely join or leave the mobile crowdsourcing system from $l_{i}$ to a maximal distance $d_{i}$. Therefore, $w_{i}$ can be shown by $w_{i}=<l_{i},v_{i},d_{i},\Omega_{i},q_{i}>$.

### 3.2. Crowdsourcing Task with Categories

In this section, we present the definition of a crowd task with categories in a mobile crowdsourcing system. Assume that $\psi =\left\{g_{1},g_{2},\ldots ,g_{o}\right\}$ is a set of all categories of tasks in the mobile crowdsourcing, that is to say, there are *o* types of tasks in the mobile crowdsourcing.

**Definition** **2.**
*(Crowdsourcing Tasks with Categories) $T=\left\{t_{1},t_{2},\ldots ,t_{n}\right\}$ denotes the crowdsourcing task set in the mobile crowdsourcing, each $t_{j}\left(1\leq j\leq n\right)$ has a category set $G_{j}=\left\{g_{1},g_{2},\ldots ,g_{s}\right\}\left(\in \psi \right)$, which means that $t_{j}$ belongs to s categories. Each task has a budget $B_{j}$ for crowd workers.*


As given in Definition 2, the task requester publishes a mobile task $t_{j}$ with the category that requires the worker to physically move to a specific location $l_{j}$ and arrive at $l_{j}$ before the arrival deadline $e_{j}$. At the same time, the task requester also specifies the budget $B_{j}$, which is the maximum amount he/she is willing to pay for the worker. This budget $B_{j}$ can be a bonus cash or bonus point in mobile crowdsourcing system. At the same time, this task belongs to certain categories, e.g., it can belong to both decoration and construction. Therefore, $t_{j}$ can be shown by $t_{j}=<l_{j},G_{j},e_{j},B_{j}>$.

### 3.3. Allocation Problem

Given the crowdsourcing task set *T*, the crowdsourcing worker set *W*, the crowdsourcing task and the crowdsourcing workers appear one after another in a certain order. The solution goal is to find the task allocation of *T* and *W*, and M⊆T×W. To maximize the total utility of task allocation, it should satisfy the following constraints:**Deadline constraint**: When a crowdsourcing task $t_{j}$ or a crowdsourcing worker $w_{i}$ appears, task allocations can be given, and the crowdsourcing worker’s arrival time satisfies the deadline of $t_{j}$.**Invariant constraint**: Once allocated, task allocations cannot be changed.**Spatial constraint**: The task allocation $<w_{i},t_{j}>$ needs to satisfy that the moving distance of $w_{j}$ cannot exceed his maximal moving distance.**Budget constraint**: The payment paid by the requester for $w_{i}$ cannot exceed the maximal budget of $t_{j}$.

To allocate worker $w_{i}$ to task $t_{j}$, we need to pay his cost $c_{ij}$ (the driving cost in this paper), which is related to the travel cost from $w_{i}$’s position $l_{i}$ to $t_{j}$’s position $l_{j}$. The value of $c_{ij}$ can be calculated by the gas/transport fee per kilometer and the moving distance. For public transportation, the cost of $c_{ij}$ can be calculated by multiplying the cost per kilometer by the distance traveled. For walking, we can also provide workers with a compensation fee that is proportional to his/her distance traveled.

Without loss of generality, assume that the cost $c_{ij}$ is proportional to the travel distance $dist(l_{i},l_{j})$, where $dist(l_{i},l_{j})$ is the distance function between $w_{i}$ and $t_{j}$. Formally, we have $c_{ij}=F_{i}\cdot dist\left(l_{i},l_{j}\right)$, where $F_{i}$ is the constant parameter (e.g., gas/transport fee per kilometer). In our paper, we use the actual distance in the Google Maps API as the distance function. The descriptions for symbols are shown in Table 2.

## 4. The Proposed Allocation Model

In this section, the corresponding solutions for the problems of time relationship strength and geographical relationship strength are presented.

### 4.1. Time Relationship Strength

The time relationship strength between workers and friends can be expressed by basic interactions between two workers (e.g., sending messages) or static public information (e.g., common hobbies). We represent this interaction or static information as an impact factor. The impact factor represents the identifiable and countable expression of the worker relationship. It can influence the relationship strength in a positive or negative way based on social attributes.

Impact factors can exist in multiple social networks. The impact factors of each social network may have different weights. The weight of impact factor indicates its relative impact on the strength of the final relationship. In our mobile crowdsourcing system, the weight of impact factor for each source is manually assigned. According to the behavior of workers, the weights of impact factors are determined. According to the evaluation for Facebook social network [40], the weight of common hobbies is set as 0.3, the weight of learning in the same school is set as 0.1, and the weight of boyfriend/girlfriend relationship is set as 0.9. The final relationship strength is also affected by the number of impact factors, because the using frequency of social networks has an impact on the strength of relationships.

People’s activities are divided into short-term activities, long-term activities and permanent activities. The impact factor of a short-term activity represents a single expression of the relationship. The date and time of this event can be recognized. In addition, we can estimate the duration of relationship through experience, e.g., when sending a message, the duration of influence time is two days.

The impact factor of a long-term activity is used for the activities that we can identify the start time and end time. As the previous type, we can estimate the duration of its impact, e.g., studying at the same school from September 2017 to June 2020, which the duration of impact is estimated as three years.

The impact factor of a permanent activity represents a permanent activity or some static information, and we cannot define its start and end dates, e.g., family relations. Therefore, the relationship strength of one impact factor is shown by Equation (Equation 1).
(1)$$I\left(s,f\right)=w_{sf}\cdot \frac{\sum_{i=1}^{u}F_{t}}{1+ln(1+u_{c})}$$
where *s* is a social network, $w_{sf}$ is the weight of impact factor *f* of social network *s*, *u* indicates the number of impact factors in the relationship between the worker and his friend, $u_{c}$ is the count of instances of impact factors in social network *s*, and $F_{t}$ indicates the function expressing time impact. Because we use the time attribute to calculate the relationship strength of one impact factor, Equation (Equation 1) is used to calculate the time relationship strength of one impact factor. $F_{t}$ depends on the type of impact factor. We determine three time impact functions based on the types of activities.

**Short-term activity:**(2)$$F_{t}(t_{sf},t_{c},t_{a})=\left\{\begin{array}{cc} \frac{1}{1+\log_{t_{sf}}\left(max\left(1,t_{c}-t_{a}\right)\right)} & t_{c}\geq t_{a}\\ 0 & t_{c}<t_{a} \end{array}\right.$$
where $t_{sf}$ is the time of the impact duration in days, $t_{c}$ indicates the current time, $t_{a}$ is the start time of the short-term activity, and $t_{c}-t_{a}$ is the time of duration of the short-term activity.

**Long-term activity**:(3)$$F_{t}(t_{sf},t_{c},t_{s},t_{e})=\left\{\begin{array}{cc} \frac{1}{1+\log_{t_{sf}}\left(max\left(1,t_{c}-t_{e}\right)\right)} & t_{c}>t_{e}\\ \frac{1}{2} & t_{c}\in <t_{s},t_{e}>\\ 0 & t_{c}<t_{s} \end{array}\right.$$
where $t_{sf}$ is the time of the impact duration in days, $t_{c}$ is the current time, $t_{s}$ indicates the start time of the long-term activity, and $t_{e}$ means the end time of the long-term activity. For example, a worker and his friend are studying at the same school from September 2017 to June 2020. $t_{sf}$ is three years, $t_{s}$ is September 2017, $t_{e}$ is June 2020, and $t_{c}$ is the current system time.

**Permanent activity**:(4)$$F_{t}=\frac{1}{2}$$

Because permanent activities cannot be estimated, it can be set as a fixed value $\frac{1}{2}$. According to the time relationship strength, its calculation method is discussed by an instance.

The relationships between people can be roughly divided into three types. Figure 2a shows the direct relationship between friends. For example, Workers A and B are friends. Figure 2b shows the indirect relationship between friends. For example, Workers A and B are not friends, but they have a common friend. Figure 2c shows the complex relationship between friends. For example, there is a direct relationship and an indirect relationship between Workers A and E.

#### 4.1.1. Direct Relationship Strength

The direct relationship between the two workers is obtained by Equation (Equation 5).
(5)$$TRS_{d}\left(w_{1},w_{2}\right)=\sum_{s}^{n}\sum_{f}^{m}I\left(s,f\right)$$
where *n* represents the number of social networks and *m* represents the number of impact factors in the social network.

#### 4.1.2. Indirect Relationship Strength

The indirect relationship strength between the two workers is obtained by Equation (Equation 6).
(6)$$TRS_{id}\left(w_{1},w_{2}\right)=e^{-\gamma d}\prod_{i=1}^{d}TRS_{d}\left(w_{1},w_{2}\right)$$
where γ represents the attenuation coefficient of the length of relationship path, and *d* represents the length of relationship strength. In addition, $e^{-\gamma d}$ is an attenuation function whose value varies continuously from 0 to 1, as illustrated in Figure 3. The x-coordinates of Figure 3 indicate the length of relationship strength *d*. $e^{-\gamma d}$ represents the weight factor of relationship path in all paths. In general, there are multiple relationship paths between two workers. Because the length of relationship has different effects on the relationship strength, we cannot consider all paths as equivalent. Therefore, we use weighted average to calculate the indirect relationship strength for all paths.

Assume there are *n* paths represented as $\left\{P_{1},P_{2},\ldots ,P_{n}\right\}$ between worker $w_{1}$ and worker $w_{2}$. The weights are $\left\{e^{-\gamma d_{1}},e^{-\gamma d_{2}},\ldots ,e^{-\gamma d_{n}}\right\}$, respectively. Therefore, the indirect relationship strength for all paths is shown by Equation (Equation 7) through improving Equation (Equation 6).
(7)$$TRS_{id}\left(w_{1},w_{2}\right)=\frac{\sum_{i=1}^{n}\left[e^{-\gamma d_{i}}\prod_{i=1}^{d_{i}}TRS_{d}\left(w_{1},w_{2}\right)\right]}{\sum_{i=1}^{n}e^{-\gamma d_{i}}}$$

#### 4.1.3. Composite Relationship Strength

Based on the direct relationship strength and indirect relationship strength, the final composite relationship strength is computed by Equation (Equation 8).
(8)$$TRS\left(w_{1},w_{2}\right)=\alpha \cdot \sum_{s}^{n}\sum_{f}^{m}I(s,f)+\beta \cdot \frac{\sum_{i=1}^{n}\left[e^{-\gamma d_{i}}\prod_{i=1}^{d_{i}}TRS_{d}\left(w_{1},w_{2}\right)\right]}{\sum_{i=1}^{n}e^{-\gamma d_{i}}}$$
where α and β are the weights of direct relationship strength and indirect relationship strength, respectively, and α+β=1.

### 4.2. Geographical Relationship Strength

In mobile crowdsourcing, the location information is an important attribute, which can affect the efficiency of task allocation. When a worker and his friend have strong time relationship strength, but the distance between them is very large, it cannot allocate the same task for them. In this paper, we use GeoHash code to calculate the location-based relationship strength, and different ranges can be obtained according to different digits. GeoHash converts two-dimensional latitude and longitude into a string. For example, Figure 4 shows the GeoHash strings of nine regions in Beijing, namely WX4ER, WX4G2, WX4G3, etc. Each string represents a rectangular area. That is to say, all points (latitude and longitude coordinates) in this rectangular area share the same GeoHash string, which can not only be used for the calculation of geographical relationship strength, but also protect the location privacies of workers and tasks in mobile crowdsourcing (only indicates the approximate location of the area rather than the specific point). The GeoHash string represents not a point, but a rectangular area, such as the code wx4g0ec19. The worker can post the address code to indicate that he/she stays near Beihai Park, but does not expose his/her exact position.

Algorithm 1 shows the calculation method of geographical relationship strength based on GeoHash. First, entering the latitude and longitude coordinates $l_{w_{1}}$ and $l_{w_{2}}$ of worker $w_{1}$ and worker $w_{2}$, the coordinate format is l=<longitude,latitude>. Then, the corresponding binary codes $geoLog_{w_{1}}$, $geoLat_{w_{1}}$, $geoLog_{w_{2}}$, and $geoLat_{w_{2}}$ of latitude and longitude of the position are obtained according to the GeoHash code. We combine the GeoHash codes of latitude and longitude to generate a new string following the rule of *even bit longitude, odd bit latitude*. The new resulting string is processed into a code sequences $Base_{w_{1}}$, $Base_{w_{2}}$ following the rules of base32 encoding. Then, the two codes are compared starting from the first position of the sequence until a different encoding is encountered to obtain $Base_{con}$.

Let $Precision_{max}$ be the maximal precision. We utilize Softmax activation function to represent their geographical relationship strength. The calculation of geographical relationship strength is shown by Equation (Equation 9).
(9)$$LRS=\frac{e^{\left|Base_{con}\right|}}{\sum_{j=1}^{Precision_{max}}e^{j}}$$
where $\left|Base_{con}\right|$ indicates the length of $Base_{con}$, LRS∈[0,1].

**Algorithm 1** The LRS algorithm.
**Require:** $l_{w_{1}}$, $l_{w_{2}}$, $Precision_{max}$;**Ensure:** LRS;
1:$geoLog_{w_{1}}$, $geoLat_{w_{1}}$, $geoLog_{w_{2}}$, $geoLat_{w_{2}}$← Calculate GeoHash binary code based on latitude and longitude;2:$geo_{w_{1}}$, $geo_{w_{2}}$← Combine $geoLog_{w_{1}}$, $geoLat_{w_{1}}$, $geoLog_{w_{2}}$, $geoLat_{w_{2}}$ with 2 strings according to the principle of “even bit longitude, odd bit latitude";3:$Base_{w_{1}}$, $Base_{w_{2}}$← Process $geo_{w_{1}}$, $geo_{w_{2}}$ according to the rules of base32 encoding;4:$Base_{con}$← Calculate the same string starting from the first digit of the two strings;5:LRS← By Equation (Equation 9);6:**return**LRS;


### 4.3. Mixed Relationship Strength

In the above two sections, the time attribute and geographical attribute are discussed. The calculation method of relationship strength between worker $w_{1}$ and friend $w_{2}$ is obtained by Equation (Equation 10).
(10)$$RS\left(w_{1},w_{2}\right)=-\ln\left(TRS\left(w_{1},w_{2}\right)\cdot LRS\left(w_{1},w_{2}\right)\right)$$
where $TRS(w_{1},w_{2})$, $LRS\left(w_{1},w_{2}\right)\in \left[0,1\right]$, we get the logarithm of $TRS\left(w_{1},w_{2}\right)\cdot LRS\left(w_{1},w_{2}\right)$ in order to amplify the data.

### 4.4. The FRS-MC Algorithm

In this section, we present the friend relationship-based task allocation model FRS-MC. Algorithm 2 shows the process of FRS-MC algorithm. The task set *T*, the worker set *W*, and the maximal utility $U_{max}$ learned through historical records are given. The tasks and workers appear randomly in mobile crowdsourcing system. The new task will be added into the set AppearTaskList. The new worker will be added into the set CandidateList, and the friends of the new worker in CandidateList will be found too. If the CandidateList set is empty, there is no worker in the allocated worker set who are friends of the new worker. By randomly selecting a threshold $e^{k}$, the utilities of tasks and workers should be greater than $e^{k}$. The value of *k* will be randomly selected from 0,1,…,θ−1, where θ can be obtained by $\left\lceil \ln(U_{max}+1)\right\rceil$. $U_{max}$ can be obtained through historical records. The value of *k* will be generated randomly according to the probability $p_{k}$. When the task allocation is completed, the algorithm updates the weight $\omega_{k}$ of threshold $e^{k}$ by the equation ωk=ωk(1+δ)ukukUmaxUmax, and updates the values of SumW and $p_{k}$, correspondingly, where $u_{k}$ indicates the utility if $e^{k}$ is always used as the threshold. If the CandidateList set is not empty, then we calculate the time relationship strength and geographical relationship strength of the newly appearing worker and his friends who already exist in mobile crowdsourcing system. The system selects the worker with the strongest relationship with the newly-emerged worker, and finds the category of task that this worker has done. Then, the system finds the existing tasks that have are in this category. The system then selects the task for them with the highest budget from this task set for allocation.

**Algorithm 2** The FRS algorithm.
**Require:** *W*,*T*,$U_{max}$;**Ensure:** *M*;
1:ListWT← mix *W*,*T*;2:shuffle ListWT;3:
$\theta \leftarrow \left\lceil \ln(U_{max}+1)\right\rceil$
4:
i=0,1,⋯,θ−1,wi=1,SumW=∑iwi,pi=wiwiSumWSumW
5:**for** each object∈ListWT
**do**6: **if**
object is worker **then**7:  CandidateList← Find a list of workers who are friends with the object in the system8:  **if**
CandidateList is not Null **then**9:   **for** each candidate∈CandidateList
**do**10:    Calculate the time relationship strength between candidate and object11:    Calculate the geographical relationship strength between candidate and object12:   **end for**13:   Get the categories of tasks that workers having the strongest relationship with object in CandidateList have performed and allocate this kind of task to object14:   Add to *M*15:  **else**16:   Randomly choose one as the k value from 0,1,⋯,θ−1 according to the probability $p_{i}$17:   Allocate tasks that meet all constraints and utilities of the allocation are higher than $e^{k}$18:   Add to *M*19:   ωk=ωk(1+δ)ukukUmaxUmax20:   Update SumW and $p_{i}$21:  **end if**22: **else**23:  Append to AppearTaskList24: **end if**25:
**end for**



## 5. Experiments and Analysis

### 5.1. Experimental Methodology

In this section, the methods of experimental evaluation and experimental results are presented. Real world (REAL) and synthetic (SYN) datasets were used to verify the performance of the proposed task allocation model.

The Yelp dataset [41] as real data was utilized to conduction comparison experiments. Yelp is a social network dedicated to local business directory services and review sites, which includes 188,593 enterprises, 1,518,169 users and 5,996,997 comments. For this dataset, we treated Yelp users as mobile crowdsourcing workers and enterprises as mobile crowdsourcing tasks. In Yelp dataset, the location information of workers is not provided. In the experiments, one of the worker’s comments was selected, and the position of the corresponding enterprise of the selected comment was considered the position of the worker. Therefore, the location attribute of workers can be obtained. Figure 5a,b show the distribution of workers and tasks provided by the Amap [42]. The average rating in this dataset was considered the worker’s service quality. In addition, the Yelp dataset also provides the rating that the company receives, which considered the budget of the task. The allocatable time for the task was set to the business hours of the enterprise.

According to the synthetic data, based on Gaussian distribution, the locations of workers and tasks in the 2D data space $\left[0,5\right]{}^{2}$ were generated. The actual latitude and longitude were used to calculate the distance. According to the quality of worker’s sensed data, we generated data that fit the Gaussian distribution with an expected value of 0.7 and a variance of 0.1. In terms of worker’s friends, we set a set of 10,000 workers in advance. For each worker, we randomly sampled 10 workers from the set as the friends of the worker. In terms of task categories, we set a set of 20 categories in advance. According to each task, we randomly sampled three categories from the set as the category of the task.

We conducted comparison experiments for GeoHash coding precision, and the precision was set as 8, 10 and 12, respectively. To evaluate the performance of our proposed algorithm, we compared our proposed FRS algorithm with greedy algorithm, random threshold algorithm and adaptive threshold algorithm, and the precision was set as 12. Greedy selects a “best” worker-and-task assignment with the highest utility each time, which is a local optimal approach. The random threshold algorithm [43] randomly selects a threshold, and when the allocation is greater than the threshold, the allocation is completed. The adaptive threshold algorithm [44] is improved on the basis of the random threshold algorithm, which can improve the probability of occurrence of a ideal threshold.

For each group of experiments, the utility, the number of allocations and the running time were compared. All experiments were run on Windows 10 operating system with Intel(R) Core(TM) i5-6500 @3.20GHz CPU, 8.00GB Memory, Python2.7 and PyCharm 2018.2.4 simulation platform.

### 5.2. Experiments on Synthetic Data

#### 5.2.1. Effect of the Precision

FRS model with different geographical precisions will have different effects on the experimental results, thus three groups of comparison experiments were conducted to illustrate the effect of different geographical precisions for utility, number of allocations and running times. Each group of comparison experiments included 10 experiments, and the experimental settings are shown in Table 3.

The x-coordinates of Figure 6a–c indicate the number of workers in the mobile crowdsourcing system.The y-coordinate of Figure 6a indicates the allocation utility, the y-coordinate of Figure 6b indicates the number of allocations, and the y-coordinate of Figure 6c indicates the running time. Figure 6a presents the experimental results on utilities under different values of precision. The utilities increased with the increase of workers, and the utilities reached steady state when the number of workers was big enough. However, the FRS algorithm when precision=12 was more efficient than the FRS algorithm when precision=8 and precision=10, and the utilities could reach the maximal value more quickly. This was because higher geographical precision led to more accurate geographical relationship strength between workers and their friends. It was easier to find a friend with a stronger relationship for the worker.

Figure 6b shows the experimental results on the number of allocations under different values of precisions. The number of allocations increased with the increase of workers, and the number of allocations reached stable state in the end. However, the FRS algorithm when precision=12 had a higher allocation utility than the FRS algorithm when precision=8 and precision=10, and reached the maximal number of allocations more quickly.

Figure 6c shows the influence of different geographical precisions on algorithm running time. In Figure 6c, it can be seen that different precisions had little effect on running time.

From these three groups of comparison experiments, it can be seen that the FRS algorithm when precision=12 was better than other FRS algorithms in terms of utilities and number of allocations. In the following experiments, we selected precision=12 for FRS algorithm.

To illustrate the applicability of our proposed FRS algorithm for different numbers of workers and tasks, the proposed FRS algorithm on synthetic dataset was compared with greedy algorithm, random threshold algorithm and adaptive threshold algorithm. According to the synthetic dataset, we conducted the comparison experiments on small-scale dataset and large-scale dataset.

#### 5.2.2. Effect of Small-Scale Data

The experimental settings for small-scale data are shown in Table 4. The x-coordinate of Figure 7a–c all indicate the number of workers in the mobile crowdsourcing system. The y-coordinate of Figure 7a indicates the allocation utility, the y-coordinate of Figure 7b indicates the number of allocations, and the y-coordinate of Figure 7c indicates the running time.

Figure 7a shows that, when the number of workers was more than 500, our proposed FRS algorithm was better than the other three algorithms and had an upward trend. When the number of workers exceeded 800, the utilities of the other three algorithms decreased, but the utility of FRS algorithm still increased steadily. This was because that FRS algorithm was designed based on the relationship of friend. The more workers there were, the easier it was to find friends of a certain worker, thus the more tasks could be allocated to this worker, and the higher was the utility of the allocation.

Figure 7b shows that the numbers of allocations of greedy, random threshold and adaptive threshold algorithms were almost same, and the number of allocations of FRS algorithm was lower than the other three algorithms. This was because that FRS algorithm needed to calculate the strength of friendship relationship to find the tasks that could be allocated. The matching conditions were more stringent, which resulted in fewer allocations. However, it did not affect the overall utility, as shown in Figure 7a.

Figure 7c shows that the running time of FRS algorithm was higher than the other three algorithms, because the FRS algorithm first finds the worker’s friend and then calculates the strength of relationship with the friend. Although the FRS algorithm took a long time to run, it took milliseconds to process each object. Therefore, the proposed FRS algorithm can meet the real-time requirements.

From these three groups of experiments, we found that our proposed FRS algorithm is suitable for mobile crowdsourcing system on small-scale dataset, and is better than the other three algorithms in terms of utility.

#### 5.2.3. Effect of Large-Scale Data

The experimental settings for large-scale data are shown in Table 5. The x-coordinate of Figure 8a–c indicates the number of workers in the mobile crowdsourcing system. The y-coordinate of Figure 8a indicates the allocation utility, the y-coordinate of Figure 8b indicates the number of allocations, and the y-coordinate of Figure 8c indicates the running time.

Figure 8a shows that, when the number of workers was more than 4000, the FRS algorithm was better than the other three algorithms and had an upward trend. In addition, it can be seen that, when the number of workers exceeded the number of tasks, the utilities of the other three algorithms remained almost unchanged, while the FRS algorithm increased steadily. This was because that FRS algorithm was designed based on the relationship of friend. The larger was the number of workers, the easier it was to find the friends of a certain worker is, thus the more tasks could be allocated to this worker, and the higher was the utility of the allocation.

Figure 8b shows that the numbers of allocations of greedy, random threshold and adaptive threshold algorithms were almost same, and the number of allocation of FRS algorithm was lower than the other three algorithms. This was because that FRS algorithm needs to calculate the strength of friendship relationship to find the tasks that can be allocated, thus the matching conditions are more stringent, which results in the decrease of the number of allocations. However, it did not affect the overall utilities, as shown in Figure 8a.

Figure 8c shows that the running time of FRS algorithm was higher than the other three algorithms. This was because that the FRS algorithm firstly finds the worker’s friend and then calculates the strength of friend relationship. Although the FRS algorithm took a long time to run, it took milliseconds to process each object. Therefore, it could meet the real-time requirements.

Through the above comparison experiments, it was found that the FRS algorithm is applicable for mobile crowdsourcing system, and is better than the other three algorithms in terms of utilities. In addition, the greater is the number of workers in the platform, the better the effect will be.

#### 5.2.4. Effect of the Number of Tasks

In addition, we conducted an experiment with a group of 1000 workers with an increasing number of tasks from 1000 to 10,000 to show the effect of different numbers of tasks on experimental results. The experimental settings are shown in Table 6. The x-coordinate of Figure 9a–c indicate the number of workers in mobile crowdsourcing system. The y-coordinate of Figure 9a indicates the allocation utility, the y-coordinate of Figure 9b indicates the number of allocations, and the y-coordinate of Figure 9c indicates the running time.

Figure 9a shows that the utilities of all algorithms increased from the beginning of the experiment. When the number of tasks was more than 2000, the utilities of the other three algorithms decreased, but the FRS algorithm continued to increase steadily. This was because, when the number of tasks increased, it was easier to choose the tasks with higher budget for allocation.

Figure 9b shows that the numbers of allocations of greedy, random threshold and adaptive threshold algorithms were almost the same, and the number of allocations of FRS algorithm was lower than the other three algorithms. This was because that FRS algorithm needs to calculate the strength of friend relationship to find the tasks that can be allocated. Therefore, the matching conditions are more stringent, which results in the decrease of the number of allocations. However, it did not affect the overall utilities, as shown in Figure 9a.

Figure 9c shows that the running time of FRS algorithm was higher than the other three algorithms because the FRS algorithm firstly finds the worker’s friend and then calculates the strength of friend relationship. Although the FRS algorithm took a long time to run, it took milliseconds to process each object. Therefore, it could meet the real-time requirements.

Through this group of experiment, it was found that, when the number of tasks in mobile crowdsourcing system was greater than the number of workers, the FRS algorithm was better than the other three algorithms in terms of utilities, and the effect was significant. From the experimental results, it was also found that the more workers and tasks there are in the mobile crowdsourcing system, the better the effect will be.

### 5.3. Experiments on Real Data

We also conducted experiments on Yelp public dataset to verify the adaptability of the proposed FRS algorithm. Yelp original data have 188,593 tasks and 1,518,169 workers, among which 3000 tasks and 50,000 workers were selected for the experiments. The experimental settings are shown in Table 7.

Figure 10a shows that FRS algorithm was significantly better than the other three algorithms in terms of utility, which was the same as the experimental results on synthetic data. The number of workers in the experiment was very large, which made it easier for the system to find the friends of workers and allocate tasks.

Figure 10b shows that the number of allocation of FRS algorithm was lower than greedy algorithm, but better than random threshold algorithm and adaptive threshold algorithm. FRS algorithm allocated tasks almost completely, thus its allocation efficiency was better than the other algorithms. According to the FRS algorithm, the more workers there are in the system, the better the allocation effect will be.

Through the experiments, we found that our proposed FRS algorithm is fully adaptable to the real world, and the more workers there are in the system, the better the effect will be.

## 6. Conclusions

With the development of mobile crowdsourcing, how to allocate appropriate tasks for workers has become the focus of research. Starting from the homogeneity of human beings, the relationship between friends in social networks is applied to mobile crowdsourcing system. This paper proposes a friend-based task allocation model (FRS), which can maximize the utility of mobile crowdsourcing system. The GeoHash coding mechanism is adopted in the process of calculating the strength of worker relationship, which effectively protects the location privacy of workers. The effectiveness and adaptation of the proposed FRS model in small-scale and large-scale mobile crowdsourcing systems were verified through comparison with traditional Greedy algorithm, Random threshold algorithm and Adaptive threshold algorithm on real dataset and simulation datasets.

In future works, the task allocation mechanism based on social attributes of workers will be further studied to improve the efficiency of the mobile crowdsourcing systems.

## Figures and Tables

**Figure 1 sensors-19-00921-f001:**
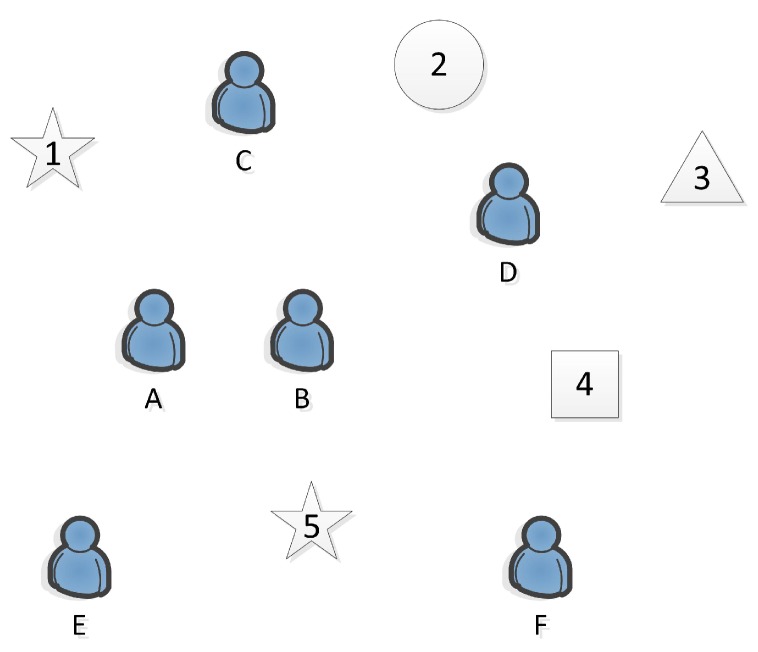
An Example of workers and tasks.

**Figure 2 sensors-19-00921-f002:**
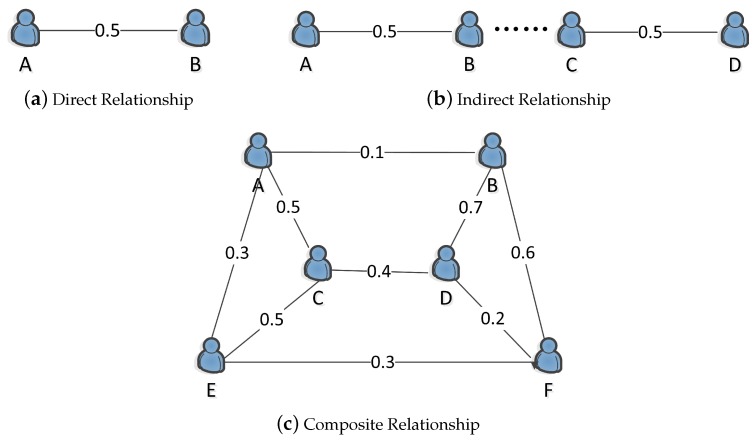
Types of relationship.

**Figure 3 sensors-19-00921-f003:**
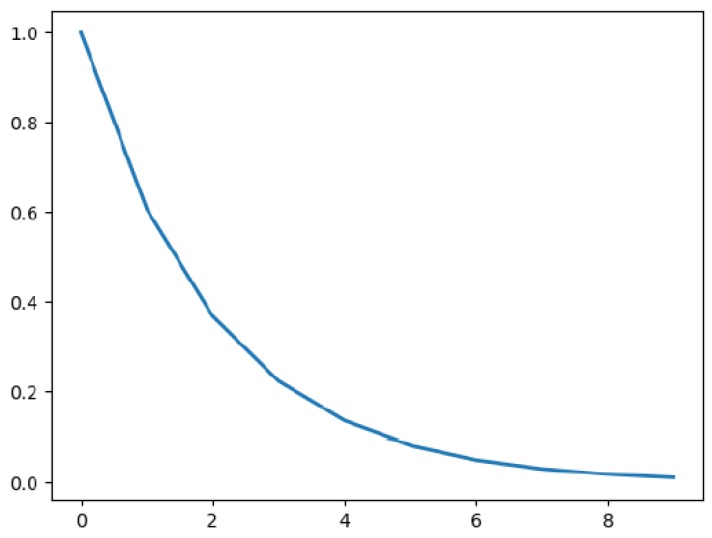
Attenuation function of $e^{-\gamma d}$.

**Figure 4 sensors-19-00921-f004:**
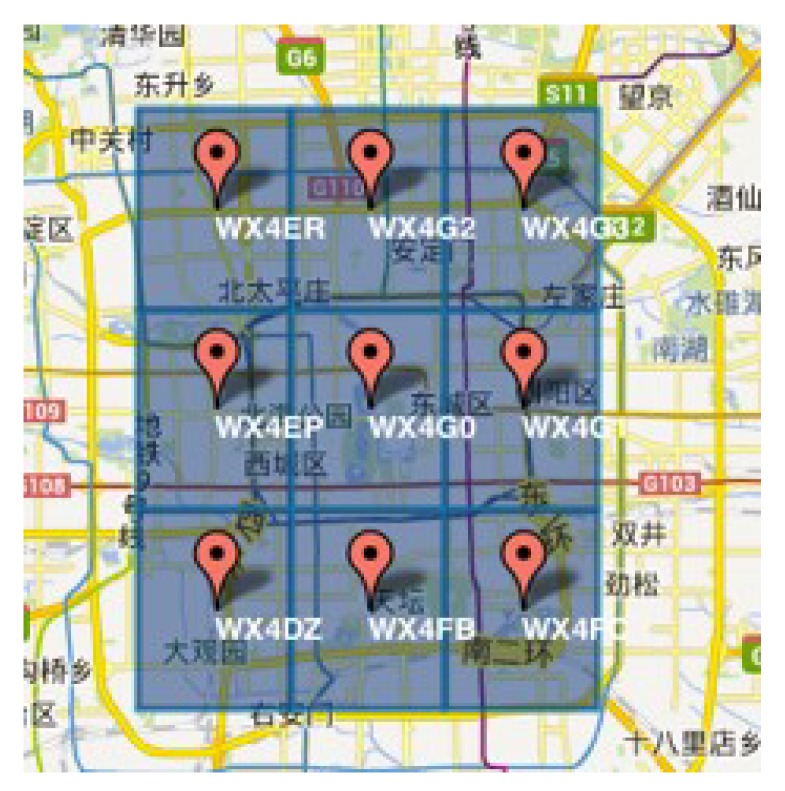
GeoHash.

**Figure 5 sensors-19-00921-f005:**
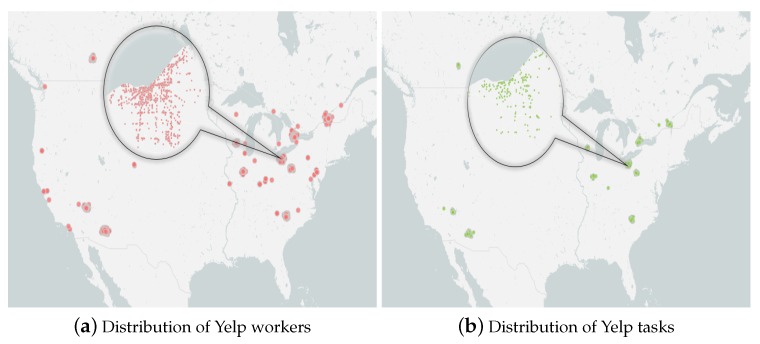
Yelp dataset.

**Figure 6 sensors-19-00921-f006:**
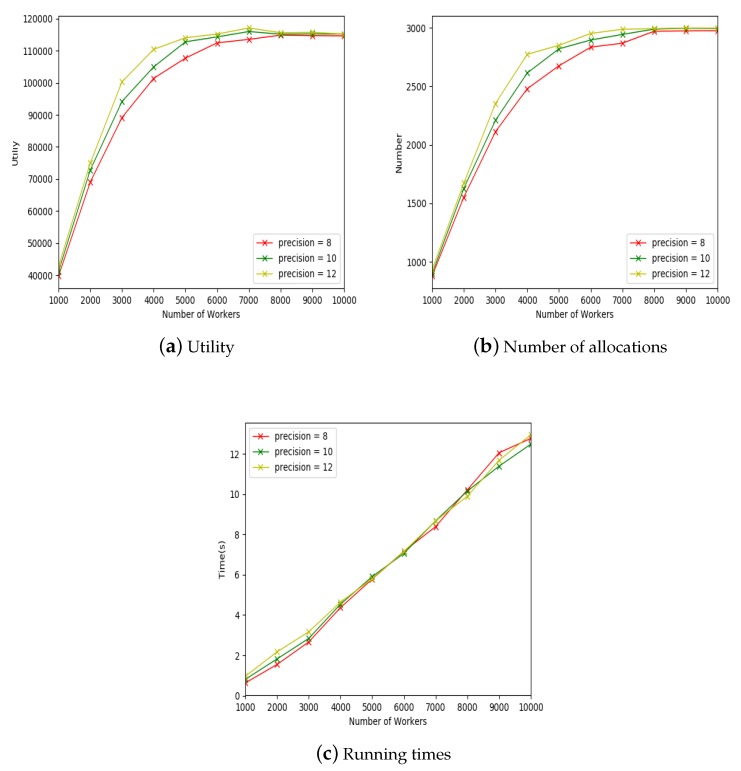
The effect of precision.

**Figure 7 sensors-19-00921-f007:**
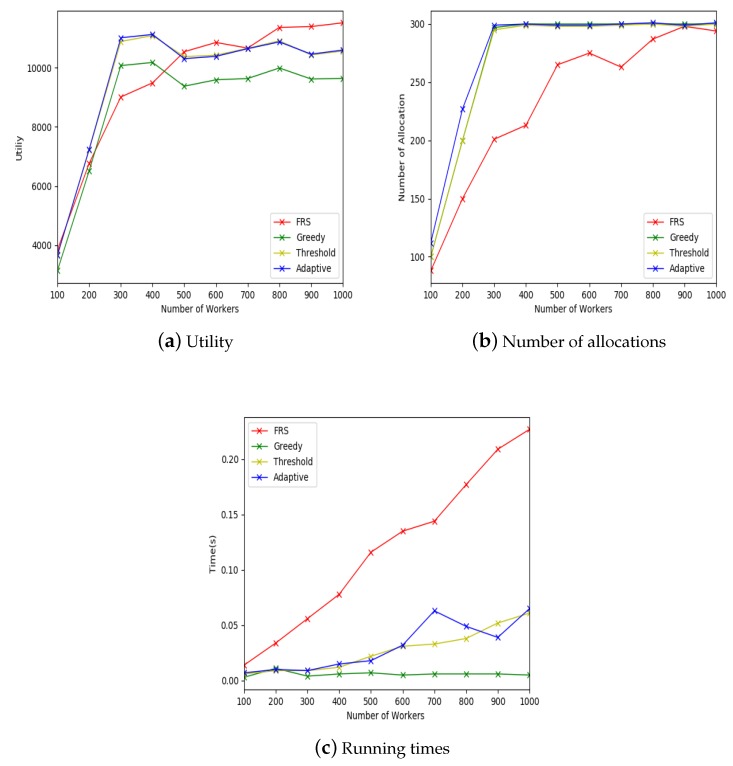
Effect of small-scale data.

**Figure 8 sensors-19-00921-f008:**
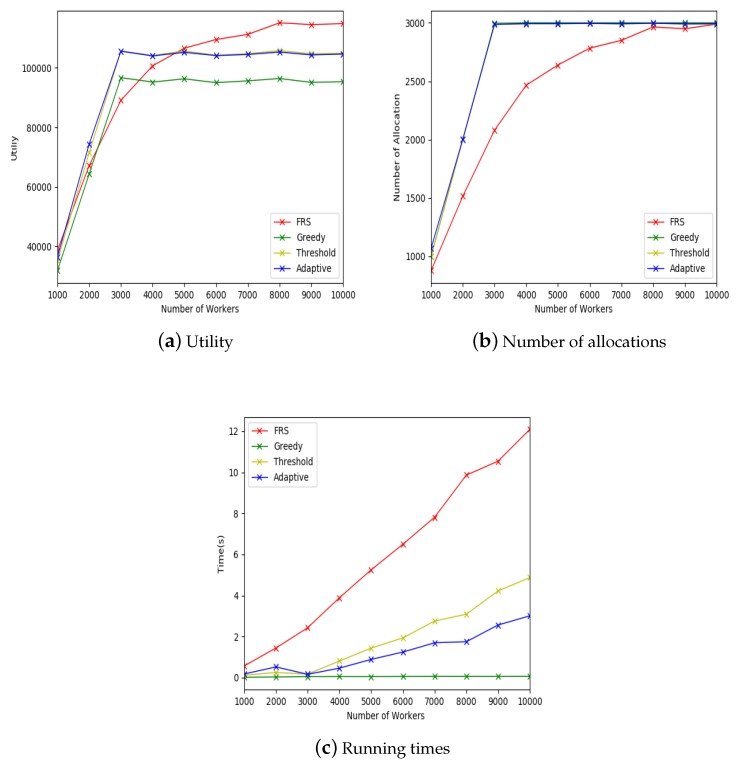
Effect of large-scale data.

**Figure 9 sensors-19-00921-f009:**
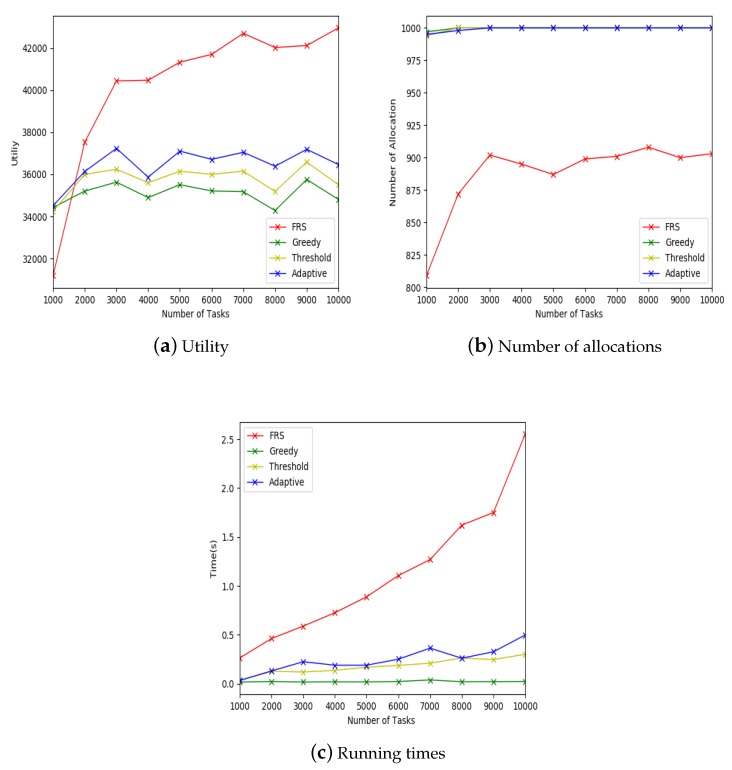
Effect of the number of tasks.

**Figure 10 sensors-19-00921-f010:**
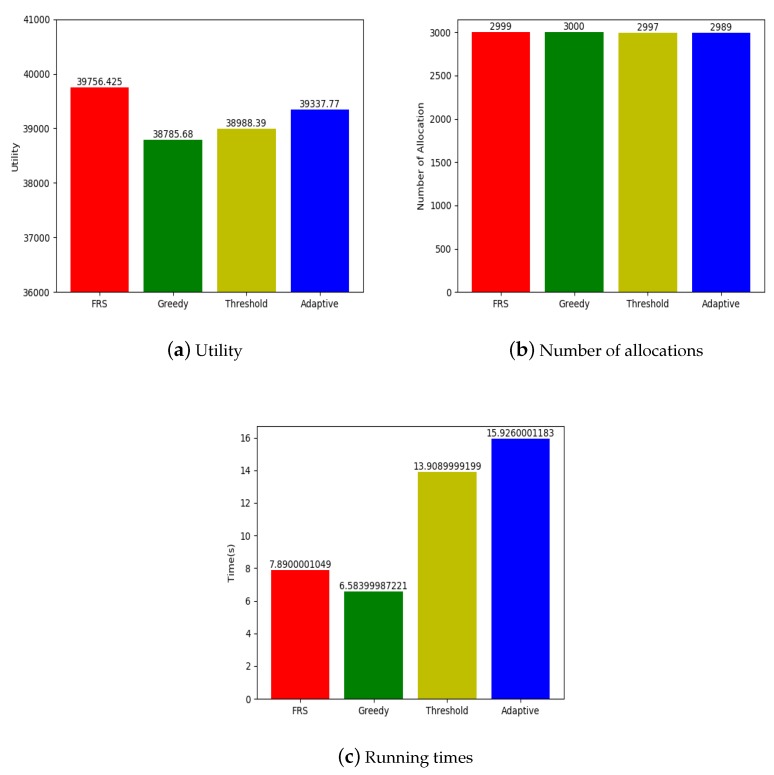
Comparison result of Yelp dataset.

**Table 1 sensors-19-00921-t001:** Task/worker categories.

Task/Worker	Catogories
A, B	c1
C, D, E, F	c2, c3
1, 5	c1
2	c2
3	c3
4	c4

**Table 2 sensors-19-00921-t002:** Symbols and notations.

Symbol	Description
$P_{t}$	the set of all workers at timestamp *t*
*W*	the set of all workers in mobile crowdsourcing
*T*	the set of all tasks in mobile crowdsourcing
$l_{i}$	the position of $w_{i}$
$l_{j}$	the position of $t_{j}$
$v_{i}$	the speed of $w_{i}$
$d_{i}$	maximum moving distance of $w_{i}$
$\Omega_{i}$	the set of $w_{i}$’s friends
$q_{i}$	the service quality of $w_{i}$
$G_{j}$	the category set of $t_{j}$
$e_{j}$	the deadline of $t_{j}$
$B_{j}$	the budget of $t_{j}$

**Table 3 sensors-19-00921-t003:** Experimental settings for the effect of precision.

Utility/Number of Allocation/Running Time										
Workers	1000	2000	3000	4000	5000	6000	7000	8000	9000	10,000
Tasks	3000	3000	3000	3000	3000	3000	3000	3000	3000	3000
Precision	8/10/12	8/10/12	8/10/12	8/10/12	8/10/12	8/10/12	8/10/12	8/10/12	8/10/12	8/10/12

**Table 4 sensors-19-00921-t004:** Experimental settings for effect of the small-scale data.

Utility/Number of Allocation/Running Time										
Workers	100	200	300	400	500	600	700	800	900	1000
Tasks	300	300	300	300	300	300	300	300	300	300
Precision	12	12	12	12	12	12	12	12	12	12

**Table 5 sensors-19-00921-t005:** Experimental settings for effect of large-scale data.

Utility/Number of Allocation/Running Time										
Workers	1000	2000	3000	4000	5000	6000	7000	8000	9000	10,000
Tasks	3000	3000	3000	3000	3000	3000	3000	3000	3000	3000
Precision	12	12	12	12	12	12	12	12	12	12

**Table 6 sensors-19-00921-t006:** Experimental settings for effect of the number of tasks.

Utility/Number of Allocation/Running Time										
Workers	1000	1000	1000	1000	1000	1000	1000	1000	1000	1000
Tasks	1000	2000	3000	4000	5000	6000	7000	8000	9000	10,000
Precision	12	12	12	12	12	12	12	12	12	12

**Table 7 sensors-19-00921-t007:** Experimental settings for Yelp real dataset.

Utility/Number of Allocation/Running Time	
Workers	50,000
Tasks	3000
Precision	12

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
