# Peer review of "Task Allocation Model Based on Worker Friend Relationship for Mobile Crowdsourcing"

_sensors, 2019, doi:10.3390/s19040921_

Round 1

Reviewer 1 Report

Overall a good work on mobile crowd sourcing. 

(1) The idea seems novel

(2) The constraints and equations are correctly formulated

(3) Results are promising.

Some explanations about the computation and storage overhead should be discussed.

Author Response

Comment 1:

Overall a good work on mobile crowd sourcing. The idea seems novel. The constraints and equations are correctly formulated. Results are promising.

Reply:

Thanks for the comments.We read a lot of literature and spent a lot of time getting the idea. And through a lot of contrast experiments, we found that the algorithm proposed is very meaningful.

Comment 2:

Some explanations about the computation and storage overhead should be discussed.

Reply:

Thanks for the comment. Although the FRS algorithm takes a long time to run, it takes milliseconds to process each object. Therefore, it can meet the real-time requirements.The details are shown in Line 449 and 450.

Reviewer 2 Report

The manuscript describes well the suggested approach.

But there is no comparison with other approaches.

At least two different approaches similar to the suggested approach can be listed.

Their performances should be compared.

Author Response

Comment 1:

The manuscript describes well the suggested approach.

Reply:

Thanks for the comment.We have conducted a lot of contrast experiments and presented a detailed description of the algorithm in terms of utilities, allocations and runtime.

Comment 2:

But there is no comparison with other approaches. At least two different approaches similar to the suggested approach can be listed. Their performances should be compared.

Reply:

Thanks for the comment. In order to evaluate the performance of our proposed algorithm, we have compared our proposed FRS algorithm with Greedy algorithm, Random threshold algorithm and Adaptive threshold algorithm.The details are shown in Section 5. Experiments And Analysis.

Reviewer 3 Report

Dear authors

The article “Task Allocation Model Based On Worker Friend Relationship For Mobile Crowdsourcing” is very interesting and show a novel approach to data analysis. Nevertheless, in my modest opinion, this work still needs some rearrangements in order to be published. Therefore I leave some suggestions about the items that seem less clear.  

Line 14 0. Introduction or 1. Introduction

Line 15 GPS and TP (Throttle position sensor?) need to be defined

Line 18 uploading traffic information, e.g. Waze https://www.waze.com/

Line 19 author forgot ones like dating/meeting apps and games (e.g. Pokémon Go)

Line 20 authors are writing about “mobile crowdsourcing for performing spatial tasks” and then state that “a new framework has emerged, and mobile crowdsourcing has enable workers to perform spatio-temporal tasks”. This phrase should be in the beginning of the paragraph (line 15).

Line 21 Reference needed for: “At the same time, the market research institute Newzoo recently released the “2018 Global Mobile Market Report"

Line 32-33 Fifure 1 appear without being referenced in the text, the first mentions is in line 64

Line 45 References needed for: “Although there are many studies on task allocation methods for mobile crowdsourcing”

Line 47  References needed as examples of : “At present, few research works…. “

Line 48 “There is a Chinese proverb called “The birds of a feather gather together" does not seem to properly link with the rest of the sentence.

Line 51 References needed for the study/results of Carolyn Parkinson

Line 63 please replace A, B, C, D, E and F by A to F and 1, 2, 3, 4, 5 by 1 to 5 (or 1-5)

Line 64 “As shown in Fig.1,worker A and worker B are university roommates who live together everyday”: Sorry but from figure 1 I cannot see that they are roommates and special that they live together everyday (can be roommates attending different classes?)

Line 69-70 Table 1 – what is the meaning of worker’s Categories?

Line 72 please replace “if workers A and worker B are classmates” by “if workers A and B are classmates”.

Line 73  Only now authors explain that workers assist to the same classes: “We assume that workers A and B are majoring in computer science and technology”: Why calling them workers and not students?

Line 78 “in this paper, we propose a Friend Relationship Strength Mobile Crowdsourcing (FRS-MC)” and again in line 84 – “In this paper, we propose the task allocation model”

Line 85 why chossing GeoHash coding? What are the advantages and disadvantages regarding other options, e.g. H3, Open Location Code (OLC), r-trees, etc.

Line 104 “With the development of various sensors (such as GPS, TP) and wireless mobile networks (such as 4G, 5G), the ability of smart devices to process data is getting more and more strong. It’s easy to participate in mobile crowdsourcing for performing spatial tasks at specific locations for people, such as taking photos/videos[1], uploading traffic information, weather conditions, and online courier picking and booking services. It has greatly improved the quality of life and promoted the development of online technology.” is equal to Line 15 “With the development of various sensors (such as GPS, TP) and wireless mobile networks (such as 4G, 5G), the ability of smart devices to process data is getting more and more strong. It’s easy to participate in spatial tasks at specific locations for people, such as taking photos/videos[1] uploading traffic information, weather conditions, online courier picking[6] and booking services. It has greatly improved the quality of life and promoted the development of online technology.”

Line 169-170  “location privacy protection mechanism CKD” is for chronic kidney disease (CKD) patients?

Line 179 Relationships among friends in social networks: This needs to be better explained: which networks? What was the methodology?

Line 195 “a category set G” in table 1 categories are denoted as C

Line 223 “we use the actual distance in the Google Maps API as the distance function”. Since Google provides the time distance shouldn’t this be a better way to joint car, public transportation and pedestrian movement in one calculation?

Line 239 the total weight exceeds 1.0 “the weight of common hobbies is set as 0.3, the weight of learning in the same school is set as 0.1, and the weight of boyfriend/girlfriend relationship is set as 0.9.”

Line 251 equation 4 is of no use

Line 264 y = e gd cannot be figure 3 title

Line 289 please replace “but a rectangular area, such as the code wx4g0ec19, which represents a rectangular area” by but a rectangular area, such as the code wx4g0ec19”

Line 201 why authors choose a “Softmax activation function”?

Line 335 please provide a reference for “Yelp dataset” How is data extracted from yeld.

Line 351 Why “20 categories”? which are they? And why those specifically?

Line 355 What is the implication of using “precision is set as 8, 10 and 12 respectively”?

Line 368 “The x-coordinates of Fig.6(a), Fig.6(b) and Fig.6(c) indicate the number of workers in the mobile crowdsourcing system. The y-coordinate of Fig.6(a) indicates the allocation utility, the y-coordinate of Fig.6(b) indicates the number of allocation, and the y-coordinate of Fig.6(c) indicates the running time.” is information for the figures axes and not for a paragraph.

Line 404 “Greedy, Random threshold and Adaptive threshold algorithms” appear here for the first time. Please succinctly define them or at least provide some references. They are 3 or 4 algorithms?

Line 482 Despite an article quality is not measured by its length, in my modest opinion seven lines is a little short for conclusion.

Keep the good work

Best regards

Round 2

Reviewer 2 Report

Comments are addressed.

Reviewer 3 Report

Dear Authors

Thank you for your responses to my suggestions, I am sorry If I was a little picky. I am fully satisfied with the changes and/or the justifications you give, so I am recommending the article to be accepted. Best regards and keep on the good work.